# Isolation and Characterization of Poeciguamerin, a Peptide with Dual Analgesic and Anti-Thrombotic Activity from the *Poecilobdella manillensis* Leech

**DOI:** 10.3390/ijms241311097

**Published:** 2023-07-04

**Authors:** Chaoming Wang, Mengrou Chen, Xiaoyu Lu, Shuo Yang, Min Yang, Yaqun Fang, Ren Lai, Zilei Duan

**Affiliations:** 1Key Laboratory of Bioactive Peptides of Yunnan Province/National & Local Joint Engineering Center of Natural Bioactive Peptides, Kunming Institute of Zoology, Chinese Academy of Sciences, Kunming 650223, China; 2University of Chinese Academy of Sciences, Beijing 100049, China; 3School of Life Sciences, Tianjin University, Tianjin 300000, China; 4KIZ-CUHK Joint Laboratory of Bioresources and Molecular Research in Common Diseases, National Resource Center for Non-Human Primates, Kunming Primate Research Center/National Research Facility for Phenotypic & Genetic Analysis of Model Animals (Primate Facility), Kunming Institute of Zoology, Chinese Academy of Sciences, Kunming 650107, China; 5Sino-African Joint Research Center, Kunming Institute of Zoology, Chinese Academy of Sciences, Kunming 650223, China

**Keywords:** poeciguamerin, *Poecilobdella manillensis*, analgesic, anti-thrombotic, thrombus formation

## Abstract

When *Poecilobdella manillensis* attacks its prey, the prey bleeds profusely but feels little pain. We and other research teams have identified several anticoagulant molecules in the saliva of *P. manillensis*, but the substance that produces the paralyzing effect in *P. manillensis* is not known. In this study, we successfully isolated, purified, and identified a serine protease inhibitor containing an antistasin-like domain from the salivary secretions of *P. manillensis*. This peptide (named poeciguamerin) significantly inhibited elastase activity and slightly inhibited FXIIa and kallikrein activity, but had no effect on FXa, trypsin, or thrombin activity. Furthermore, poeciguamerin exhibited analgesic activity in the foot-licking and tail-withdrawal mouse models and anticoagulant activity in the FeCl_3_-induced carotid artery thrombosis mouse model. In this study, poeciguamerin was found to be a promising elastase inhibitor with potent analgesic and antithrombotic activity for the inhibition of pain and thrombosis after surgery or in inflammatory conditions.

## 1. Introduction

The phenomenon of pain arises from three distinct sources. Firstly, pain can elicit a response or feedback from a living organism toward an environmental stimulus. Secondly, pain can arise in response to tissue damage or infection, triggering activation of the immune system. Lastly, pain can stem from abnormalities within the nervous system [1]. Among them, one group found an important role for leukocyte elastase and SerpinA 3 N, a member of the serine protease inhibitor superfamily, in the regulation of abnormal neurological pain [2,3,4]. Neutrophil elastase cleavage of protease-activated receptor-2 (PAR 2) at Ala^66^↓Ser^67^ and Ser^67^↓Val^68^ mediates cAMP accumulation and extracellular regulated protein kinases 1/2 (ERK 1/2) activation, and induction of PAR 2 coupled to Gαs leads to PAR 2 and transient receptor potential vanilloid 4 (TRPV 4)-mediated extracellular Ca^2+^ influx, and blockade of neutrophil elastase, PAR 2, and ERK 1/2 activity reduces inflammation and pain [5,6,7].

Venomous organisms have undergone extensive evolution, resulting in the development of unique zootoxins that serve crucial roles in survival, sustenance, and defense [8,9,10,11,12,13]. These animal toxins harbor a diverse array of peptides with multifunctional and practical properties, including peptides with antibacterial [14,15], antithrombotic [16,17], analgesic [18,19], antioxidant [20,21], antihypertensive [22,23], neuroactive [24,25], and regulatory effects [26,27]. Animal toxins are a treasure trove for drug development. These natural bioactive peptides have garnered increasing attention given their potential in clinical applications, such as therapies for thrombosis, hypertension, bacterial infection, and abnormal pain.

Several peptides with bioactivity have been identified in leeches. For example, the sylvestin peptide (43 amino acids long, molecular weight 4790.4 Da), isolated and purified from the forest leech (*Haemadipsa sylvestris*), can prolong activated partial thromboplastin time (APTT) without affecting prothrombin time (PT) by inhibiting the biological activities of kallikrein and blood coagulation factor XIIa (FXIIa), thus showing potential in the treatment of ischemic stroke [28]. Similarly, the HSTX-I peptide, isolated from salivary gland secretions of *H. sylvestris*, can exert a significant analgesic function by inhibiting voltage-gated sodium (NaV) channels (NaV1.8 and NaV1.9) [29,30]. In addition, the neuropeptide Y-HS (36 amino acids long), isolated and purified from *H. sylvestris*, contains an amidated C-terminus and exhibits significant inhibitory effects on the production of inflammatory cytokines, such as tumor necrosis factor-α (TNF-α), interferon-γ (IFN-γ), interleukin-6 (IL-6), and monocyte chemotactic protein-1 (MCP-1) [31]. In addition, the novel elastase inhibitor HMEI-A, also isolated and purified from *P. manillensis*, can inhibit the formation of neutrophil extracellular traps (NETs) [32].

In the current study, we isolated, purified, and characterized an analgesic and antithrombotic peptide (named poeciguamerin) derived from salivary gland secretions of the *Poecilobdella manillensis* leech species. The poeciguamerin peptide was shown to contain an antistasin-like domain, serving as a serine protease inhibitor [33]. Remarkably, poeciguamerin significantly inhibited elastase activity and slightly inhibited FXIIa and kallikrein activity, but exhibited no inhibitory effects on FXa, trypsin, or thrombin activity. Furthermore, poeciguamerin not only exhibited pain inhibition by targeting elastase but also displayed inhibitory effects on coagulation activity through suppression of the intrinsic coagulation pathway involving FXIIa and kallikrein. Thus, during its predatory attacks, *P. manillensis* utilizes the poeciguamerin peptide present in its saliva to obtain blood from its prey while simultaneously shielding itself from detection.

## 2. Results

### 2.1. Isolation and Purification of Poeciguamerin

Lyophilized powder of *P. manillensis* salivary gland secretions was resuspended by dissolution in phosphate buffer (0.1 M PB, pH 6.0). As shown in Figure 1A, the salivary gland secretions were separated using a Sephadex G-50 gel filtration column, resulting in the detection of three distinct peaks. We obtained three peaks in this step, each representing a protein mixture. We then determined that the 2# peak (indicated by arrows in Figure 1A) in this step had significant inhibition of elastase activity by detecting its effect on elastase activity (Figure 1B). Next, the 2# peak in Figure 1A was further separated by reverse-phase high-performance liquid chromatography (RP-HPLC) using a C8 column (Figure 1C). In this step, we obtained a total of 10 peaks, each representing a protein mixture (Figure 1C). We again found that the 6# peak (indicated by arrows in Figure 1C) in this step has a significant inhibition of elastase activity by testing its effect on elastase activity (Figure 1D). Finally, the 6# peak in Figure 1C was further separated on a Mono S^TM^ 5/50 GL column coupled to an AKTA Explorer 10S fast protein liquid chromatography (FPLC) system (Figure 1E). In this step, we obtained a total of four proteins (Figure 1E). In this step, we again found that peaks 1# and 2# significantly inhibited elastase activity by testing their effect on elastase activity (Figure 1F). However, due to the low content of protein represented by the 1# peak in Figure 1E (indicated by arrows) in *P. manillensis* salivary gland secretion, it was not possible to obtain sufficient protein for the next experiments. Therefore, we temporarily focused our attention on the 2# peak in Figure 1E for the next experiment and named it “poeciguamerin”.

### 2.2. Primary Structure of Poeciguamerin

All Elution Peak 2, which inhibited elastase activity, was collected and subjected to lyophilization (indicated by arrows in Figure 1E). Using the Edman degradation method (Appendix A) and local blast with our transcriptome data of the salivary glands of *P. manillensis*, we determined that the complete mature polypeptide of poeciguamerin contained 57 amino acids with the sequence “VDEKAEVTDDLCGDKTCSGAQVCQNDACVCSPVRCMIMCPNGFKLDENGCEYPCSCA” (Figure 2A). Sequence comparison revealed that poeciguamerin is very similar to gelin, also an elastase inhibitor previously identified from *P. manillensis*, but only a partial sequence of gelin was identified and the complete sequence was not [34] (Figure 2B). In addition, our peptide shared approximately 72% homology with guamerin, including a highly conserved cysteine position, with both belonging to the same class of proteins possessing antistasin-like domains (Figure 2B). Finally, we determined the molecular weight and purity of poeciguamerin using matrix-assisted laser desorption/ionization time-of-flight mass spectrometry (MALDI-TOF-MS). The molecular weight of the natural poeciguamerin protein was 6058.6 Da (Figure 2C).

### 2.3. Effect of Poeciguamerin on Protease and Coagulation

As poeciguamerin can act as a serine protease inhibitor and possesses an antistasin-like domain, it may also exhibit anticoagulant properties. Thus, we examined the effects of poeciguamerin on proteases such as FXIIa, kallikrein, trypsin, thrombin, FXa, and elastase using appropriate chromogenic substrates. As shown in Figure 3A, poeciguamerin exhibited a dose-dependent inhibition of elastase activity and slight inhibition of FXIIa (Figure 3B) and kallikrein (Figure 3C) activities but showed no inhibitory effects on the activities of FXa (Figure 3D), trypsin (Figure 3E), or thrombin (Figure 3F). The inhibition constant (*Ki*) values of poeciguamerin for elastase, FXIIa, and kallikrein were 262.4 nM (Figure 4A), 10.72 μM (Figure 4B), and 18.97 μM (Figure 4C), respectively. Thus, poeciguamerin may have relatively potent analgesic activity but relatively weak anticoagulant activity.

### 2.4. Analgesic Activity of Poeciguamerin

To verify the functionality of poeciguamerin, we employed a thiogalactoside (IPTG)-induced prokaryotic expression system to express the poeciguamerin fusion protein with a His tag (Appendix A). Following enterokinase digestion of the His tag (Appendix A), we purified the enzymatically cleaved poeciguamerin using HPLC (Appendix A). The inhibitory effects on elastase enzymatic activity were evaluated among the three collected peaks, with the second protein peak demonstrating significant inhibition (Appendix A). Subsequently, using MALDI-TOF MS, we determined that the linear molecular weight of the protein was 6118 Da, corresponding to one molecule of glycine minus one molecule of water compared to the natural molecule (Appendix A). Thus, the expressed active protein demonstrated consistency with the natural activity and the correct molecular weight, thereby confirming the successful in vitro expression of poeciguamerin.

To evaluate the biological activity of poeciguamerin, we used the foot-licking mouse model and the tail-withdrawal mouse model [37]. In the foot-licking mouse model, we placed the mice in a hot plate apparatus at a temperature of 55 ± 0.5 °C. The pain thresholds of the mice were measured twice every 15 min and hind paw licking was used as the observation parameter. Female mice showing a pain response within 10–30 s were selected and randomly divided into four groups. In the control group, 25 μL elastase (6 mg/kg) was injected into the base of the right hind paw, while in the experimental group, poeciguamerin (0, 6, 12 mg/kg) was injected intravenously 10 min before the elastase injection. Morphine (0.2 mg/kg, i.v.) was used as a positive control. The cumulative number of foot licks during the first phase (0–10 min) (Figure 5A) and the second phase (10–30 min) (Figure 5B) was recorded. The results showed that plantar injection of elastase elicited a painful response in the mice, whereas poeciguamerin attenuated this painful response. In the withdrawal mouse model, 20 mice with a tail withdrawal latency of 4–8 s were selected for this experiment, with 5 mice in each group. Poeciguamerin (0, 6, and 12 mg/kg) or morphine (0.2 mg/kg) was injected into the tail vein of the mice before stimulation of the tail with a hot water bath for 10 min. The tails of the mice were stretched in hot water at 45 °C and the time of tail flicking was recorded. The results showed that hot water bath stimulation elicited a pain response in the mice, whereas poeciguamerin attenuated this pain response (Figure 5C). These results confirm the analgesic activity of poeciguamerin and highlight its ability to alleviate pain induced by elastase and thermal stimulation.

### 2.5. Effects of Poeciguamerin on FeCl_3_-Induced Carotid Artery Injury

To determine the impact of poeciguamerin on coagulation, we used a mouse model of FeCl_3_-induced carotid artery injury to assess the effects of poeciguamerin on thrombus formation. Different concentrations of recombinant poeciguamerin (0, 1.25, 2.5 mg/kg) and sodium heparin (20 mg/kg) were administered through the tail vein 10 min prior to the surgical procedure to establish the model. After exposure to the mouse carotid artery, blood flow was monitored for 0, 4, 8, 12, 16, 20, 24, and 28 min after applying 10% FeCl_3_ to the artery. As shown in Figure 6A,B, poeciguamerin inhibited thrombus formation in a dose-dependent manner, exhibiting a similar trend to that observed with sodium heparin.

## 3. Discussion

Many venomous organisms utilize analgesic peptides within their venom to hinder the production of pain in their prey to aid in feeding, hunting, and parasitization. Several analgesic peptides have been isolated and characterized. For example, peptide presynaptic antagonists (omega-agatoxins) purified from the venom of the funnel-web spider (*Agelenopsis aperta*) can block neuromuscular transmission at the presynapses and inhibit pain signaling in insects [38]. Omega-phonetoxin IIA peptides derived from *Phoneutria nigriventer* venom can inhibit high-threshold voltage-gated calcium currents in neurons to block pain signaling [39]. Omega-conotoxins extracted from the venom of fish-eating cone snails can selectively inhibit N-type calcium channels for severe pain control [40]. In our study, we isolated, purified, and identified a novel peptide (poeciguamerin) from the salivary gland secretions of *P. manillensis*, which was 57 amino acids in length and a molecular weight of 6058.6 Da. The peptide significantly inhibited elastase activity and slightly inhibited FXIIa and kallikrein activity, but had no inhibitory effect on blood coagulation factor Xa (FXa), trypsin, or thrombin. Remarkably, poeciguamerin demonstrated both analgesic and anticoagulant activities, allowing it to inhibit pain mediated by elastase and FeCl_3_-induced carotid artery thrombosis.

In preliminary experiments, we observed that salivary gland secretions from *P. manillensis* exhibited inhibitory effects on the bioactivity of both FXIIa and elastase [16]. Based on this finding, we isolated and purified the secretions using Sephadex G-50 dextran gel filtration chromatography, RT-HPLC, and FPLC, coupled with an elastase chromogenic substrate assay (Figure 1). Subsequently, based on the Edman degradation method (Appendix A), local blast with our transcriptome data of the salivary glands of *P. manillensis*, and MALDI-TOF-MS, the amino acid sequence of the peptide (6058.6 Da) that inhibited elastase activity was identified as “VDEKAEVTDDLCGDKTCSGAQVCQNDACVCSPVRCMIMCPNGFKLDENGCEYPC-SCA” (Figure 2A,C). Additionally, comparative analysis of the primary structure of the active peptide revealed its similarity to a class of protease inhibitors possessing an antistasin-like domain (Figure 2B).

To ensure their survival, leeches possess a diverse array of active peptides within their salivary gland secretions, which serve various analgesic, anesthetic, anti-inflammatory, vasodilatory, and anticoagulant functions, allowing for undetected and continuous feeding on the host’s blood [41]. Our results indicated that poeciguamerin significantly inhibited elastase activity (Figure 3A), with a *Ki* value of 262.4 nM (Figure 4). Although the inhibitory ability of poeciguamerin on elastase was lower compared to elastase inhibitors such as guamerin [18], HMEI-A [34], and ShSPI [42] (*ki*s are 0.08.1 pM, 16.9 nM, and 12.61 nM, respectively), compared to elastase inhibitors such as poecistasin [16] and α1-PI [43] (*Ki*s are 546.7 nM and 76 μM, etc., respectively) were more strongly inhibited. In addition, poeciguamerin significantly inhibited elastase activity (Figure 3A) and slightly inhibited FXIIa and kallikrein activity, but had no significant effect on FXa, trypsin, or thrombin activities (Figure 3B–E). Among them, the *Ki* values of poeciguamerin for FXIIa and kallikrein were 10.72 μM and 18.97 μM, respectively (Figure 4). Thus, poeciguamerin not only exhibited potent analgesic activity but also weak anticoagulant activity. This was verified in mice, whereby the peptide effectively inhibited both elastase or hot water bath-induced pain (Figure 5) and FeCl_3_-induced carotid artery thrombosis (Figure 6). While elastase requires high concentrations to inhibit the biological enzyme activities of FXIIa and kallikrein, elastase requires low concentrations to inhibit carotid artery thrombosis in a mouse model. This may be because elastase can effectively regulate platelet function and induce thrombosis [44,45,46,47].

In our previous study, we identified another bioactive peptide, poecistasin (48 amino acids long), in the salivary gland secretions of *P. manillensis*, which exhibited antithrombotic effects by inhibiting the biological activities of FXIIa, kallikrein [16]. In addition, although the analgesic effects of poecistasin have not been studied, it may also have some analgesic effect due to its ability to inhibit elastase activity. Thus, the presence of bioactive peptides, such as poecistasin and poeciguamerin, in their salivary secretions allows *P. manillensis* to inhibit host pain to avoid detection and inhibit thrombosis to enhance blood collection efficiency. Moreover, since biological preparations are more difficult to obtain, we solved this challenge by constructing prokaryotic expression vectors capable of producing large amounts of biologically active poeciguamerin. Poeciguamerin, which was found to have potent analgesic and antithrombotic activity in this study, may be used to inhibit pain and thrombosis after surgery or in inflammatory conditions.

## 4. Materials and Methods

### 4.1. Collection of Salivary Gland Secretions from P. manillensis

Leeches (*P. manillensis*) were purchased from Nanning City, Guangxi Province, China. A multifunctional electromagnetic massager (6 V, 2.5–345 Hz) was applied firmly to both corners of the mouth of a living leech for 5 to 10 s of stimulation. A colorless, viscous fluid emerged from the mouth, which was collected by repeated rinsing with salivary-stimulating solution (150 mM NaCl, 1 mM L-arginine). The supernatant was collected after centrifugation (4 °C, 12,000× *g*, 1 h), lyophilized at low pressure, and stored at −80 °C.

### 4.2. Isolation and Purification of Poeciguamerin

The lyophilized sample was dissolved in 0.1 M PB (Na_2_HPO_4_-NaH_2_PO_4_, pH 6.0) and filtered through a 0.22-μm filter. Approximately 5 mL of the resulting solution (about 200 mg of protein) was loaded onto a Sephadex G-50 gel filtration column (100 × 2.6 cm, GE Health) pre-equilibrated with the same buffer. The column was run according to the provided instructions, with elution performed using the same PB. A fully automated partial collector was used to collect samples at a flow rate of 0.3 mL/min, with one tube collected every 10 min. Absorbance at 280 nm and 215 nm was measured for alternate tubes using a spectrophotometer. The components of each peak were collected based on absorbance values, and their activity was assessed. Subsequently, the collected fractions were lyophilized under low pressure and stored for further analysis.

The lyophilized sample containing the protein peak exhibiting elastase activity inhibition was dissolved in double-distilled water, followed by centrifugation (4 °C, 12,000× *g*, 20 min) The supernatant was filtered through a 0.22-μm filter and subjected to purification using liquid chromatography. An RP-HPLC C8 column (Sepax C8, 30 × 0.46 cm) was used, with a protein loading volume of approximately 0.2 mg per sample. The elution system consisted of liquid A (double-distilled water containing 0.1% trifluoroacetic acid) and liquid B (acetonitrile containing 0.1% trifluoroacetic acid) at a flow rate of 0.7 mL/min. For gradient elution, the liquid B flow rate was increased by 1% per min. The light absorption values of the sample were detected at 280/215 nm using a visible/UV detector, with each absorption peak sample collected as a unit. The corresponding peaks were collected and subsequently lyophilized. The obtained peaks were then resuspended in solvent A (20 mM MES, pH 6.0) and further purified on a Mono S^TM^ 5/50 GL column (GE, Chicago, IL, USA) connected to an AKTA Explorer 10S FPLC system.

### 4.3. Molecular Weight and Amino Acid Sequence Determination of Poeciguamerin

The molecular weight and amino acid sequence were determined using MALDI-TOF, the Edman degradation method, and local blast with our transcriptome data of the salivary glands of *P. manillensis* [16].

### 4.4. Enzyme Kinetics

We mixed poeciguamerin with specific chromogenic substrates and proteases to evaluate the effect of poeciguamerin on various enzymatic activities. These enzymes and substrates included FXIIa (10 nM, 1212a, Enzyme Research Laboratories, South Bend, IN, USA; substrate: 0.2 mM, S-2302, Chromogenix, Milan, Italy), kallikrein (80 nM, 1303, Enzyme Research Laboratories, South Bend, IN, USA; substrate: 0.2 mM, S-2302, Chromogenix, Milan, Italy), trypsin (0.8 μM, T2600000, Sigma-Aldrich, St. Louis, MO, USA; substrate: 0.2 mM, S-2222, Chromogenix, Milan, Italy), thrombin (10 nM, 1002a, Enzyme Research Laboratories, South Bend, IN, USA; substrate: 0.2 mM, S-2238, Chromogenix, Milan, Italy), FXa (20 nM, 1011, Enzyme Research Laboratories, South Bend, IN, USA; substrate: 0.2 mM, S-2222, Chromogenix, Milan, Italy), elastase (0.4 μM, E8140, Sigma-Aldrich, St. Louis, MO, USA; substrate: 0.2 mM, 324696, Sigma-Aldrich, St. Louis, MO, USA), and other proteases were affected. The proteases and different concentrations of natural poeciguamerin (0, 0.5, 1, 2, 10, 20, 40 μM) were incubated in a total volume of 60 μL in buffer (50 mM Tris-HCl pH 7.4) for 5 min at room temperature. Subsequently, a buffer containing the appropriate enzyme substrate was added to a total volume of 100 μL. Using an enzyme marker, GEN CHS v1.09 software detected OD405 at 30-s intervals and recorded for 10 min to calculate enzyme kinetics.

### 4.5. Calculation of Inhibition Constant (Ki) in Enzyme Kinetics

The enzymatic reactions influenced by poeciguamerin were selected for further analysis, and the reaction rates at different substrate concentrations were measured using chromogenic substrates. The *Ki* values of poeciguamerin were calculated using Dixon plot curves [36], with the inhibitor concentration plotted on the horizontal axis, the inverse of the reaction rate plotted on the vertical axis, and the negative value of the inhibition constant of the inhibitor on the enzyme obtained at the intersection of the extension lines. In the Dixon plot curve analysis, the concentrations of chromogenic substrates and poeciguamerin served as variables. The reaction rate (measured as OD405/min) was determined over a 3-min interval. The concentration of poeciguamerin (0, 10, 20, 40 μM) was plotted on the horizontal axis, while the inverse of the reaction rate was plotted on the vertical axis. The negative value of the inhibition constant was obtained at the intersection point of the two-line extensions on the horizontal axis. An intersection point above the horizontal coordinate indicated competitive inhibition, while a point below indicated mixed inhibition.

### 4.6. Analgesic Activity Test

Foot-licking mouse model: As in previous experiments [37], control mice received a 25-μL injection of elastase (6 mg/kg) in the sole of the right hind paw, while experimental mice were intravenously injected with poeciguamerin (0, 6, 12 mg/kg) 10 min prior to elastase administration. Morphine (0.2 mg/kg, i.v.) was used as a positive control. There were five mice in each group. The total number of paw licks by the mice was recorded for 0–10 min and 20–30 min.

Tail withdrawal mouse model: A tail withdrawal mouse model was used to determine the effect of poeciguamerin on the response time to thermal stimulation according to previous experiments [37]. A thermostatic water bath (HH. S11-1, Shanghai, China) was used to measure tail withdrawal latency, i.e., the time required to withdraw the tail from hot water. A total of 20 mice with a tail withdrawal latency of 4–8 s were selected for this experiment, with 5 mice in each group. Before tail stimulation by a hot water bath for 10 min, poeciguamerin (0, 6, and 12 mg/kg) or morphine (0.2 mg/kg) were injected into the tail vein of the mice. The tails of the mice were stretched in the hot water at 45 °C and the time of the tail flick was recorded. The maximum time is 15 s to prevent burns.

### 4.7. FeCl_3_-Induced Carotid Artery Injury Model

Female C57BL/6J mice (23–25 g) were anesthetized as described previously [48], with body temperature maintained at 37 °C throughout the procedure. Different concentrations of recombinant poeciguamerin (1.25 and 2.5 mg/kg) and sodium heparin (20 mg/kg) were administered via tail vein injection 10 min prior to surgery. Saline was administered as a negative control. The carotid artery was dissected and separated from the attached tissue and vagus nerve. Filter paper (2 × 2 mm) pre-saturated with 10% (*w*/*v*) FeCl_3_ solution was applied to the exposed artery to induce thrombosis. Carotid blood flow was measured at 0, 4, 8, 12, 16, 20, 24, and 28 min after FeCl_3_ induction using laser scatter perfusion imaging (PeriCam PSI, HR, Stockholm, Sweden). Perfusion units were recorded in the region of interest (ROI). In this experiment, a total of 12 mice were used, 3 in each group. Moreover, the experiment was independently repeated at least twice.

### 4.8. Animals and Ethics Statement

All animal experiments were approved by the Animal Protection and Utilization Committee of the Kunming Animal Research Institute (Accreditation Code: IACUC-RE-2023-05-014). Animal experiments were performed in accordance with the NIH Guide for the Care and Use of Laboratory Animals. The female C57BL/6J mice (8 weeks, 20–23 g) were purchased from Vital River Laboratory Animal Company (Beijing, China) and housed in a sterile breeding room. All mice were maintained on a 12-h light/12-h dark cycle at 24 °C.

### 4.9. Statistical Analysis

All data obtained from independent experiments are expressed as mean ± standard error of the mean (SEM). All statistical analyses were two-tailed with 95% confidence intervals (CI). Unpaired *t*-tests were performed, and the data were analyzed using GraphPad Prism v9 and Origin 2021 software. Statistically significant differences were determined at *p* < 0.05.

## 5. Conclusions

Our study revealed a bioactive peptide, poeciguamerin, with analgesic and anti-thrombotic activity isolated from the salivary gland secretion of *P. manillensis*. First, we isolated poeciguamerin, a peptide that inhibits elastase activity, from the salivary gland secretion of *P. manillensis*, and then analyzed its primary structure and biological activity. We found that poeciguamerin is a peptide with a molecular weight of 6058.6 Da that significantly inhibits the activity of elastase and weakly inhibits the activity of FXIIa and kallikrein. On this basis, we identified poeciguamerin as a bioactive peptide with an analgesic and anti-thrombotic activity using a mouse paroxysmal activity assay and a FeCl_3_-induced carotid artery injury model. Overall, our study provides further elucidation of *P. manillensis* venom during hunting. Further studies may help in the development of novel bioactive peptide drugs to target pain and thrombosis development after surgery or in inflammatory conditions.

## Figures and Tables

**Figure 1 ijms-24-11097-f001:**
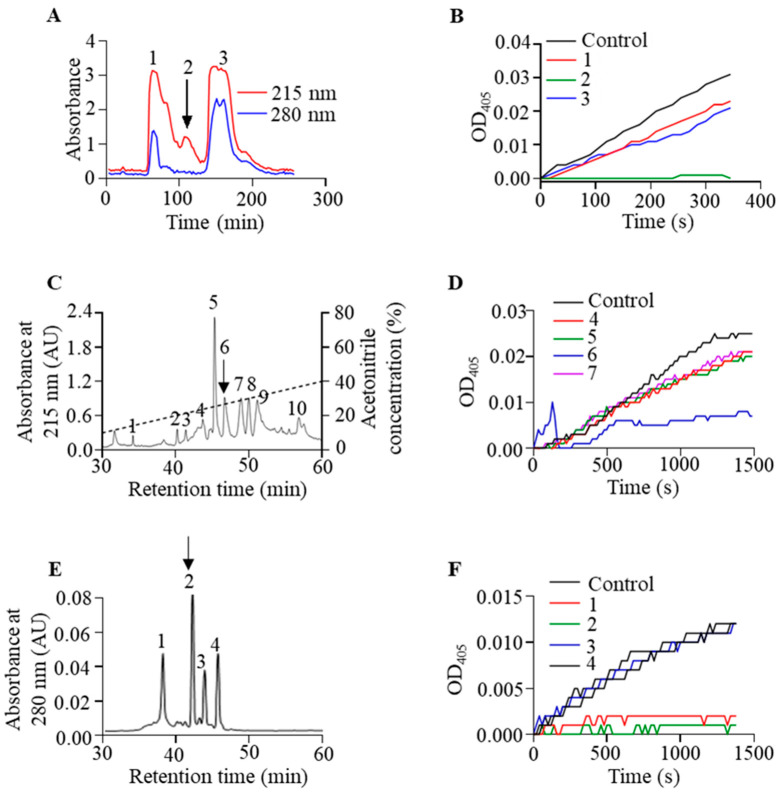
Isolation and purification of poeciguamerin from *P. manillensis* salivary gland secretions. (**A**) A Sephadex G-50 dextran gel filtration column connected to a fully automated partial collector was used for sample collection, with a flow rate of 0.3 mL/min and one tube collected every 10 min. Sample absorbance at 280 nm and 215 nm was measured in alternate tubes using a spectrophotometer. The components of each peak were collected and integrated based on absorbance values. (**B**) Elastase activity inhibition by the three peak fractions was detected. (**C**) The protein peak with elastase activity inhibition was separated and purified using liquid chromatography. An RP-HPLC C8 column with a UV detector was used to detect the light absorbance of the sample at 280 nm and 215 nm. The dashed line indicates a linear gradient of acetonitrile from 10% to 40% over 30 min. (**D**) Elastase activity inhibition was detected for the four peak components. (**E**) Collected peaks were further purified by monitoring at 280 nm using a Mono S^TM^ 5/50 GL column connected to an AKTA FPLC system. (**F**) Elastase activity inhibition was detected for the four peak fractions.

**Figure 2 ijms-24-11097-f002:**
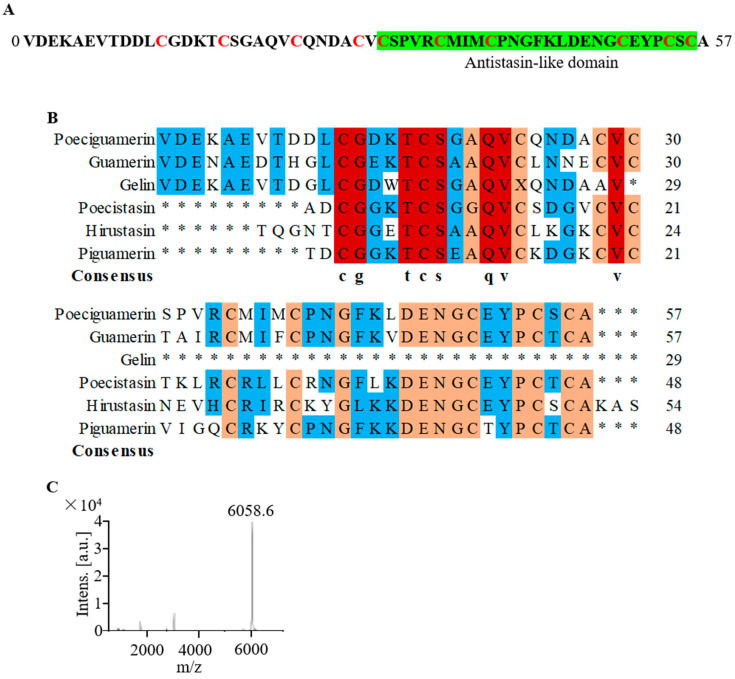
Primary structure of poeciguamerin. (**A**) Determination of amino acid sequence of poeciguamerin by the Edman degradation method and local blast with our transcriptome data (data not shown in the text) of the salivary glands of *P. manillensis*. C (Cysteine) is marked in red, and the green part represents the predicted Antistasin-like domain. (**B**) Amino acid sequences of different proteins, including poeciguamerin (*P. manillensis*), guamerin [18] (*Hirudo nipponia*, P46443), gelin [34] (*P. manillensis*, AAB27871.1), poecistasin [16] (*P. manillensis*), hirustasin [35] (*P. manillensis*, P80302), and *piguamerin* [17] (*P. manillensis*, P81499), were compared for multiple sequence alignment. Shaded areas indicate conserved sequences. Shaded areas indicate conserved sequences. The degree of amino acid sequence conservation is marked by color, and the colors from low to high are blue, orange, and red, respectively. * represents the absence of amino acids in that position. (**C**) Protein size analysis of poeciguamerin by MALDI-TOF-MS.

**Figure 3 ijms-24-11097-f003:**
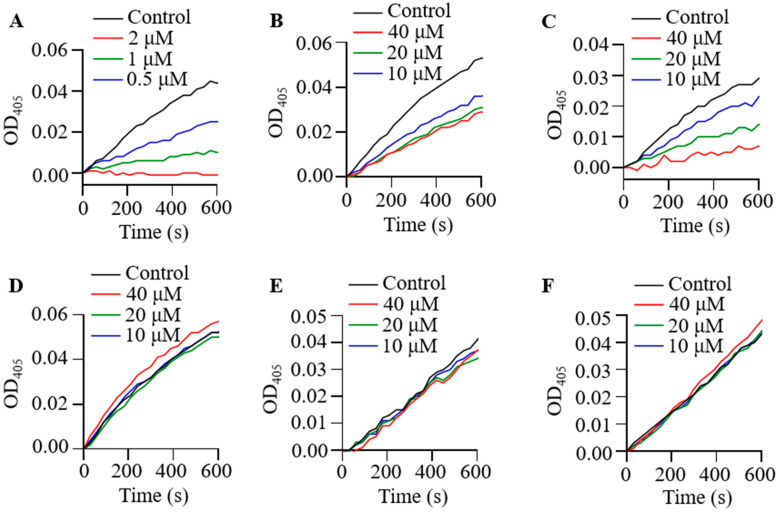
Effects of poeciguamerin on elastase and coagulation. Effects of natural poeciguamerin on elastase (**A**), FXIIa (**B**), kallikrein (**C**), FXa (**D**), trypsin (**E**), and thrombin (**F**) enzymatic activities.

**Figure 4 ijms-24-11097-f004:**
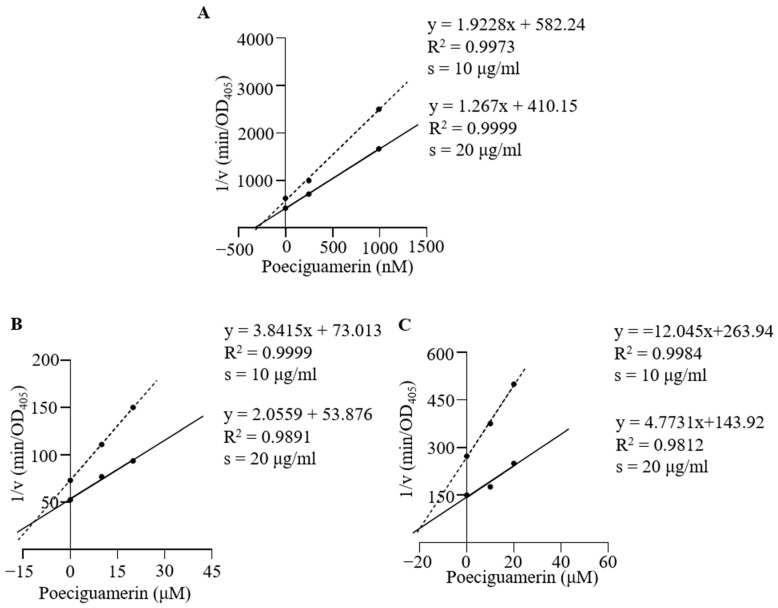
*Ki* values for the inhibition of elastase (**A**), FXIIa (**B**), and kallikrein (**C**) enzymatic activities by poeciguamerin, calculated using Dixon plot curves [36].

**Figure 5 ijms-24-11097-f005:**
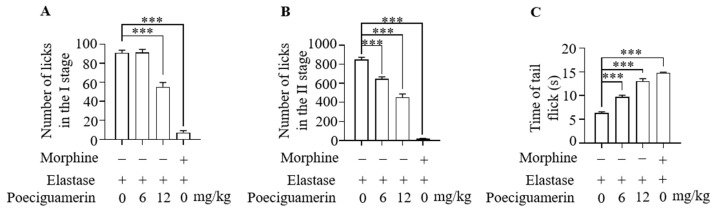
Effects of poeciguamerin on elastase or hot water bath-induced pain. Control mice were injected with 25 μL of elastase (6 mg/kg) in the sole of the right hind paw, while experimental mice were intravenously injected with poeciguamerin (12 mg/kg) 10 min prior to the elastase injection. The total time mice spent licking the paw was recorded during the I phase (0–10 min) (**A**) and the II phase (10–30 min) (**B**). *n* = 5 per group. Data are expressed as mean ± SEM and were subjected to an unpaired *t*-test, *** *p* < 0.001. In total, 20 mice with tail retraction latencies of 4 to 8 s were selected, 5 mice per group. Mice were injected with poeciguamerin (0, 6, and 12 mg/kg) or morphine (0.2 mg/kg) into the tail vein prior to stimulation of the tail with a hot water bath for 10 min. The tails of the mice were stretched in hot water at 45 °C and the time of the tail flick was recorded (**C**). *n* = 5 per group. Data are expressed as mean ± SEM and were subjected to an unpaired *t*-test, *** *p* < 0.001.

**Figure 6 ijms-24-11097-f006:**
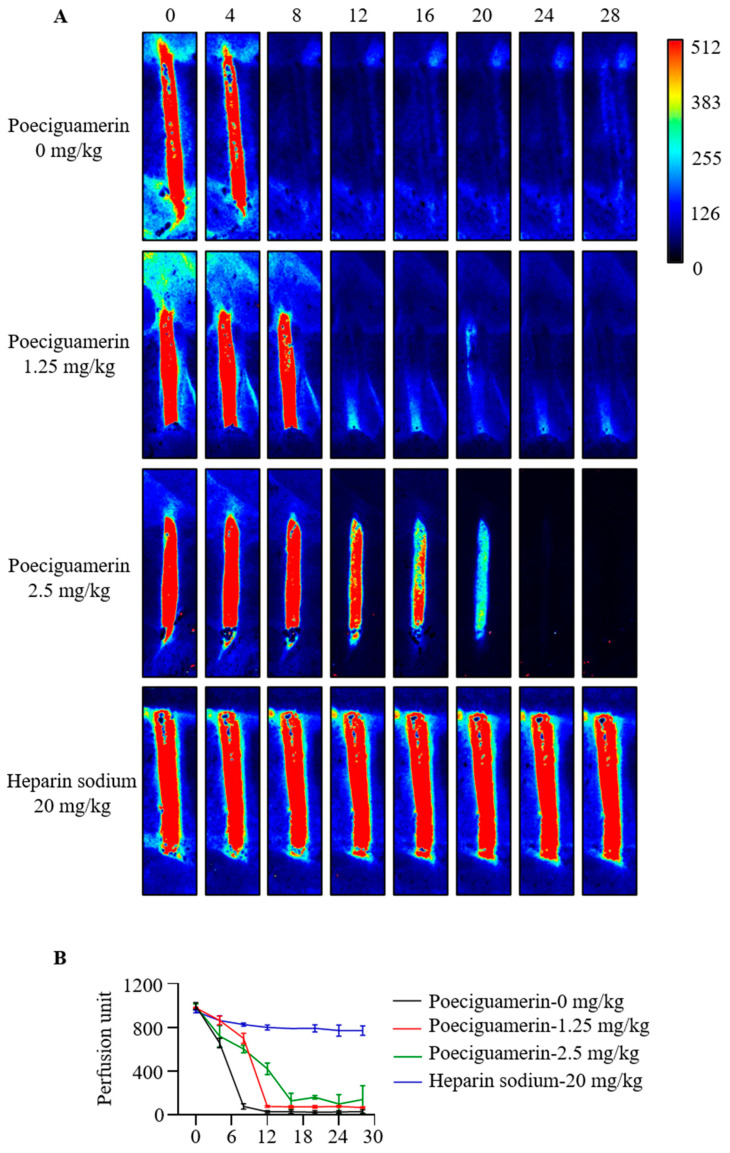
Effects of poeciguamerin on thrombosis in a mouse model of FeCl_3_-induced carotid artery injury. Different concentrations of recombinant poeciguamerin (0, 1.25, 2.5 mg/kg) and sodium heparin (20 mg/kg) were injected into the tail vein 10 min prior to the surgical procedure to establish the model. After exposure of the carotid artery, blood flow was monitored for 0, 4, 8, 12, 16, 20, 24, and 28 min after applying 10% FeCl_3_ to the exposed artery. (**A**) Image of carotid flow from laser scatter perfusion imaging. (**B**) Quantitative carotid flow data of the perfusion unit at the ROI. *n* = 3 per group. Data are expressed as mean ± SEM.

## Data Availability

For original data, please contact duanzilei@mail.kiz.ac.cn (Z.D.) and rlai@mail.kiz.ac.cn (R.L.).

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
