# Peer review of "Isolation and Characterization of Poeciguamerin, a Peptide with Dual Analgesic and Anti-Thrombotic Activity from the Poecilobdella manillensis Leech"

_ijms, 2023, doi:10.3390/ijms241311097_

Round 1
Reviewer 1 Report
1. Analgesia: Please provide on what basis did the Authors choose 12 mg/kg of the compound to be administered to mice. In line with this, please state whether the Authors determined ED50? if yes, please include the results in the manuscript. Otherwise, there is no info on the reason to choose the way (i.v.) and the dose of the compound tested. Also, please provide the number of animals used in this assay.
2. Overall this way to present analgesia properties of the compound is quite odd. The animals should be treated with various concentration of the compound, and the specific test should be used in order to determine its activity. In line with this, The Authors stated that the compound may induce pain-relieving effect. However, there is no strong evidence on that. Although the mice were observed for their behavior (licking the paw), however, no analgesic test was conducted. Therefore, in my opinion, to present the compound as a pain-killer, the Authors should provide with PWT or tail-flick or other test that may show its efficacy, either peripheral no central.
3. Please provide the chromatogram of the total compounds collected in the sample of leeches
4. Please provide the total numer of animals used and a total numer per group (subsection 4.7)
5. Subsection4.7.: The Authors stated that the compound was administered i.v. (via tail vein) 10 min prior to surgery. It would be nice of the Authors could provide the results for the compound in the aspect of its ability to bid with plasma proteins. Otherwise, please state on what basis did the Authors choose to inject the compound at this specific time-point
moderate changes are required
Author Response
Point 1. Analgesia: Please provide on what basis did the Authors choose 12 mg/kg of the compound to be administered to mice. In line with this, please state whether the Authors determined ED50? if yes, please include the results in the manuscript. Otherwise, there is no info on the reason to choose the way (i.v.) and the dose of the compound tested. Also, please provide the number of animals used in this assay.
Response 1: Many thanks for your review. We have made changes to the results, discussion, and methods. In our experiments, we identified poeciguamerin as a potentially biologically active peptide with both analgesic and anticoagulant properties. This is very similar to previous work from our laboratory [1-6]. Based on our previous work and the effects of poeciguamerin, we formulated the injection method and dose of the compound in our mouse experiments. Furthermore, we adjusted the drug concentration gradient and added a positive drug morphine group and repeated the analgesia experiment (foot licking mouse model and tail withdrawal mouse model). For example, in the foot-licking mouse model, poeciguamerin (0, 6, 12 mg/kg) was injected intravenously 10 min before 25-μl elastase stimulation of the mouse foot. Its positive reagent morphine (0.2 mg/kg) was administered in the same manner. In the tail withdrawal mouse model, poeciguamerin (0, 6, 12 mg/kg) was injected intravenously 10 min before stimulation of the mouse tail in a hot water bath. Its positive reagent morphine (0.2 mg/kg) was administered in the same manner. A total of 20 mice were used, 5 in each group. In the FeCl3-induced carotid artery injury model, poeciguamerin (0, 1.25, 2.5 mg/kg) and sodium heparin (20 mg/kg) were injected through the tail vein 10 min before surgery. A total of 12 mice were used, 3 in each group.
Reference:
- Liao, Z.; Tang, X.; Chen, W.; Jiang, X.; Chen, Z.; He, K.; Li, Q.; Duan, Z.; He, X.; Kamau, P. M.; Lv, L.; Zhang, Z.; Rong, M.; Lv, Q.; Lai, R., Shrew's venom quickly causes circulation disorder, analgesia and hypokinesia. Cell Mol Life Sci 2022, 79, (1), 35.
- Tang, X.; Chen, M.; Duan, Z.; Mwangi, J.; Li, P.; Lai, R., Isolation and Characterization of Poecistasin, an Anti-Thrombotic Antistasin-Type Serine Protease Inhibitor from Leech Poecilobdella manillensis. Toxins (Basel) 2018, 10, (11).
- Liu, W. H.; Chen, Y.; Bai, X. W.; Yao, H. M.; Zhang, X. G.; Yan, X. W.; Lai, R., Identification and characterization of a novel neuropeptide (neuropeptide Y-HS) from leech salivary gland of Haemadipsa sylvestris. Chin J Nat Med 2016, 14, (9), 677-682.
- Luan, N.; Zhao, Q.; Duan, Z.; Ji, M.; Xing, M.; Zhu, T.; Mwangi, J.; Rong, M.; Liu, J.; Lai, R., Identification and Characterization of ShSPI, a Kazal-Type Elastase Inhibitor from the Venom of Scolopendra Hainanum. Toxins (Basel) 2019, 11, (12).
- Xu, K. H.; Zhou, M.; Wu, F. L.; Tang, X. P.; Lu, Q. M.; Lai, R.; Long, C. B., Identification and characterization of a novel elastase inhibitor from Hirudinaria manillensis. Chin J Nat Med 2021, 19, (7), 540-544.
- Zhang, Z.; Shen, C.; Fang, M.; Han, Y.; Long, C.; Liu, W.; Yang, M.; Liu, M.; Zhang, D.; Cao, Q.; Chen, X.; Fang, Y.; Lu, Q.; Hou, Z.; Li, Y.; Liu, Z.; Lei, X.; Ni, H.; Lai, R., Novel contact-kinin inhibitor sylvestin targets thromboinflammation and ameliorates ischemic stroke. Cell Mol Life Sci 2022, 79, (5), 240.
Point 2. Overall this way to present analgesia properties of the compound is quite odd. The animals should be treated with various concentration of the compound, and the specific test should be used in order to determine its activity. In line with this, The Authors stated that the compound may induce pain-relieving effect. However, there is no strong evidence on that. Although the mice were observed for their behavior (licking the paw), however, no analgesic test was conducted. Therefore, in my opinion, to present the compound as a pain-killer, the Authors should provide with PWT or tail-flick or another test that may show its efficacy, either peripheral no central.
Response 2: Thank you very much for your suggestion. We have made changes in the results, discussion, and methods. Furthermore, we added the tail withdrawal mouse model to the original experiment to test the analgesic effect of poeciguamerin. At the same time, we added poeciguamerin effect concentrations - 0, 6, 12 mg/kg in the foot-licking mouse model. And morphine (0.2 mg/kg) was chosen as a positive control. Results of these experiments were added in the revised manuscript.
Point 3. Please provide the chromatogram of the total compounds collected in the sample of leeches.
Response 3: Many thanks for your review. Figure 1A in the manuscript is the chromatogram of the preliminary separation experiments of the total compounds from the salivary gland secretions of P. manillensis.
Point 4. Please provide the total numer of animals used and a total numer per group (subsection 4.7).
Response 4: Thank you very much for your suggestion. We have made changes and added the number of animals used in this text and a total number per group in the revised manuscript.
Point 5. Subsection4.7.: The Authors stated that the compound was administered i.v. (via tail vein) 10 min prior to surgery. It would be nice of the Authors could provide the results for the compound in the aspect of its ability to bid with plasma proteins. Otherwise, please state on what basis did the Authors choose to inject the compound at this specific time-point.
Response 5: Thank you for your review. We have made changes to the method. Our decision to administer poeciguamerin intravenously 10 minutes before the procedure was adjusted based on previous experience in our laboratory [1-6] and the effects poeciguamerin showed during the experiment.
Reference:
- Liao Z, Tang X, Chen W, Jiang X, Chen Z, He K, Li Q, Duan Z, He X, Kamau PM, et al. Shrew's venom quickly causes circulation disorder, analgesia and hypokinesia. Cell Mol Life Sci. 2022;79:35. doi: 10.1007/s00018-021-04116-x
- Tang X, Chen M, Duan Z, Mwangi J, Li P, Lai R. Isolation and Characterization of Poecistasin, an Anti-Thrombotic Antistasin-Type Serine Protease Inhibitor from Leech Poecilobdella manillensis. Toxins (Basel). 2018;10. doi: 10.3390/toxins10110429
- Liu WH, Chen Y, Bai XW, Yao HM, Zhang XG, Yan XW, Lai R. Identification and characterization of a novel neuropeptide (neuropeptide Y-HS) from leech salivary gland of Haemadipsa sylvestris. Chin J Nat Med. 2016;14:677-682. doi: 10.1016/S1875-5364(16)30080-2
- Luan N, Zhao Q, Duan Z, Ji M, Xing M, Zhu T, Mwangi J, Rong M, Liu J, Lai R. Identification and Characterization of ShSPI, a Kazal-Type Elastase Inhibitor from the Venom of Scolopendra Hainanum. Toxins (Basel). 2019;11. doi: 10.3390/toxins11120708
- Xu KH, Zhou M, Wu FL, Tang XP, Lu QM, Lai R, Long CB. Identification and characterization of a novel elastase inhibitor from Hirudinaria manillensis. Chin J Nat Med. 2021;19:540-544. doi: 10.1016/S1875-5364(21)60054-7
- Zhang Z, Shen C, Fang M, Han Y, Long C, Liu W, Yang M, Liu M, Zhang D, Cao Q, et al. Novel contact-kinin inhibitor sylvestin targets thromboinflammation and ameliorates ischemic stroke. Cell Mol Life Sci. 2022;79:240. doi: 10.1007/s00018-022-04257-7
Reviewer 2 Report
Authors claim that they obtain four peak fractions with Elastase activity inhibition activity at the end but they do not show any analytical evidence of the pure isolated peaks. Also fig. 1F to me seems to show that only 2 out of 4 fractions are active. Identity of the actives, 1 and 2 should be clarified.
Possibly they can show the chromatogram of Figure 2. (A) Determination of amino acid sequence of poeciguamerin by LC-MS/MS.
At some lines they are blank spaces (ex: lines 221 or 222....).
Author Response
Point 1. Authors claim that they obtain four peak fractions with Elastase activity inhibition activity at the end but they do not show any analytical evidence of the pure isolated peaks. Also fig. 1F to me seems to show that only 2 out of 4 fractions are active. Identity of the actives, 1 and 2 should be clarified.
Response 1: Thank you for your review. We have made changes to the results. As shown in Figure 1A, the salivary gland secretions were separated using a Sephadex G-50 gel filtration column, resulting in the detection of three distinct peaks. We obtained 3 peaks in this step, each representing a protein mixture. We then determined that the 2# peak in this step had a significant inhibition of elastase activity by detecting its effect on elastase activity (indicated by arrows in Figure 1A and 1B). Next, the 2# peak in Figure 1A was further separated by reverse-phase high-performance liquid chromatography (RP-HPLC) using a C8 column (Figure 1C). In this step we obtained a total of 10 peaks, each representing a protein mixture (Figure 1C). We again found that the 6# peak in this step has a significant inhibition of elastase activity by testing its effect on elastase activity (indicated by arrows in Figure 1C and 1D). Finally, the 6# peak in Figure 1C was further separated on a Mono STM 5/50 GL column coupled to an AKTA explorer 10S fast protein liquid chromatography (FPLC) system (Figure 1E). In this step, we obtained a total of 4 proteins (Figure 1E). In this step, we again found that peaks 1# and 2# significantly inhibited elastase activity by testing their effect on elastase activity (Figure 1F). However, due to the low content of protein represented by the 1# peak in Figure 1E in P. manillensis salivary gland secretion, it was not possible to obtain sufficient protein for the next experiments. Therefore, we temporarily focused our attention on the 2# peak in Figure 1E for the next experiment and named it as "poeciguamerin".
Point 2. Possibly they can show the chromatogram of Figure 2. (A) Determination of amino acid sequence of poeciguamerin by LC-MS/MS.
Response 2: Thank you for your review, which pointed out an error in the method description. We have made changes to the results, discussion, and methods. In our experiments, we first determined the amino acid sequence of poeciguamerin by LC-MS/MS method. However, because the sequence of this peptide is too long, it decays during the detection process, resulting in our inability to measure the full-length amino acid sequence of the peptide. Therefore, the only amino acid sequence we ultimately measured was "VDEKAEVTDDLCGDKTCS". After that, we searched the transcriptome data (data not shown in the text) of the salivary glands of P. manillensis. Luckily, we identified an amino acid sequence as "VDEKAEVTDDLCGDKTCSGAQ-VCQNDACVCSPVRCMIMCPNGFKLDENGCEYPCSCA", which is identical to the N-terminal sequence we determined using the Edman degradation method. Furthermore, the molecular weight of this sequence was the same as the size of poeciguamerin we obtained by MALDI-TOF (Figure 2C). mass spectra by LC-MS/MS are shown in Appendix.
Point 3. At some lines they are blank spaces (ex: lines 221 or 222....).
Response 3: Thank you very much for your suggestion. We have revised this part of the issue in the revised draft.

Round 2
Reviewer 1 Report
the manuscript has been changed and improved, therefore, in my opinion, it is suitable to be published
English is fine